# Adjunctive Cilostazol to Dual Antiplatelet Therapy to Enhance Mobilization of Endothelial Progenitor Cell in Patients with Acute Myocardial Infarction: A Randomized, Placebo-Controlled EPISODE Trial

**DOI:** 10.3390/jcm9061678

**Published:** 2020-06-01

**Authors:** Yongwhi Park, Jin Hyun Kim, Tae Ho Kim, Jin-Sin Koh, Seok-Jae Hwang, Jin-Yong Hwang, Young-Hoon Jeong

**Affiliations:** 1Department of Internal Medicine, Gyeongsang National University, School of Medicine, Jinju 52828, Korea; angio2000@hanmail.net (Y.P.); kjs0175@gmail.com (J.-S.K.); edison71@hanmail.net (S.-J.H.); jyhwang@gnu.ac.kr (J.-Y.H.); 2Cardiovascular Center, Gyeongsang National University Changwon Hospital, Changwon 51472, Korea; 3Institute of the Health Sciences, Gyeongsang National University, Jinju 52727, Korea; ajini7044@hanmail.net; 4Biomedical Research Institute, Gyeongsang National University Hospital, Jinju 52727, Korea; kth860828@gmail.com; 5Department of Internal Medicine, Gyeongsang National University, School of Medicine and Gyeongsang National University Hospital, Jinju 52727, Korea

**Keywords:** endothelial progenitor cell, cilostazol, platelet, myocardial infarction

## Abstract

Background: Endothelial progenitor cells (EPCs) have the potential to protect against atherothrombotic event occurrences. There are no data to evaluate the impact of cilostazol on EPC levels in high-risk patients. Methods: We conducted a randomized, double-blind, placebo-controlled trial to assess the effect of adjunctive cilostazol on EPC mobilization and platelet reactivity in patients with acute myocardial infarction (AMI). Before discharge, patients undergoing percutaneous coronary intervention (PCI) were randomly assigned to receive cilostazol SR capsule (200-mg) a day (*n* = 30) or placebo (*n* = 30) on top of dual antiplatelet therapy (DAPT) with clopidogrel and aspirin. Before randomization (baseline) and at 30-day follow-up, circulating EPC levels were analyzed using flow cytometry and hemostatic measurements were evaluated by VerifyNow and thromboelastography assays. The primary endpoint was the relative change in EPC levels between baseline and 30-day. Results: At baseline, there were similar levels of EPC counts between treatments, whereas patients with cilostazol showed higher levels of EPC counts compared with placebo after 30 days. Cilostazol versus placebo treatment displayed significantly higher changes in EPC levels between baseline and follow-up (ΔCD133^+^/KDR^+^: difference 216%, 95% confidence interval (CI) 44~388%, *p* = 0.015; ΔCD34^+^/KDR^+^: difference 183%, 95% CI 25~342%, *p* = 0.024). At 30-day follow-up, platelet reactivity was lower in the cilostazol group compared with the placebo group (130 ± 45 versus 169 ± 62 P2Y12 Reaction Unit, *p* = 0.009). However, there were no significant correlations between the changes of EPC levels and platelet reactivity. Conclusion: Adjunctive cilostazol on top of clopidogrel and aspirin versus DAPT alone is associated with increased EPC mobilization and decreased platelet reactivity in AMI patients, suggesting its pleiotropic effects against atherothrombotic events (NCT04407312).

## 1. Introduction

Timely reperfusion with percutaneous coronary intervention (PCI) significantly reduced mortality after acute myocardial infarction (AMI) [1]. After successful PCI procedures, appropriate inhibition of platelet reactivity with dual antiplatelet therapy (DAPT) comprised of a potent P2Y_12_ inhibitor and aspirin has been a linchpin treatment regimen to reduce stent-related and stent-unrelated ischemic events such as cerebrovascular events [2]. However, the risk of atherothrombotic events still remains relatively high even with standard DAPT [3,4].

PCI per se incurs endothelial denudation and mechanical injuries on coronary vasculature, which may result in subsequent stent-related ischemic events [5]. In addition, vascular injuries provoke activation and aggregation of platelets and thrombus formation, which culminates in acute or subacute stent thrombosis [5]. This highlights the importance of an adequate re-endothelialization and a restoration of endothelial integrity after a vascular injury. Indeed, delayed or impaired arterial healing is associated with increased risk of stent-related adverse events [6]. This prompts additional treatment options to promote a vascular healing process.

The circulating endothelial progenitor cells (EPCs) activity has been suggested as an important determinant of ischemic events in high-risk patients with cardiovascular disease [7,8]. Vascular repair after endothelial damage is mediated not only by the migration of adjacent endothelial cells but also by the bone marrow-derived EPC [5]. EPCs have emerged as an important component of response to vascular injury, which can accelerate vascular repair through rapid re-endothelialization [9,10]. Although there have been numerous therapeutic approaches to mobilize EPCs or to facilitate homing to the site of vascular injury, the results mostly have been disappointing [8,11].

Platelets play a pivotal role in vascular regeneration through paracrine effects or interactions with other blood cells [7,8,12,13]. Antiplatelet agents have therapeutic potentials to increase EPC levels and subsequently improve re-endothelialization following stent implantation [13,14,15,16,17]. Potent platelet inhibition with ticagrelor increased the number of EPC in patients with acute coronary syndrome (ACS) compared with clopidogrel, reflecting its potential effects on endothelial regeneration and prevention of stent-related adverse events [14,15,17]. Ticagrelor also increased EPC counts and decreased inflammatory cytokines in diabetic patients compared with prasugrel [17]. EPC-mobilizing effect of ticagrelor might be mediated by enhancement of plasma adenosine concentration beyond its antiplatelet effect [15,17]. Although the guidelines recommended ticagrelor or prasugrel in ACS patients, clinical data from East Asians have shown significantly increased risk of serious bleeding without definite benefit in ischemic risk among ACS patients [18,19].

Cilostazol is a dual inhibitor of phosphodiesterase 3 and adenosine reuptake [20]. Adjunctive cilostazol on top of aspirin and clopidogrel was an effective therapeutic option to overcome the risk of high on-clopidogrel platelet reactivity [21]. This regimen reduced the risks of atherothrombotic complications and revascularization in high-risk patients such as AMI and diabetes, without a trade-off of bleeding risk [22,23,24]. In an animal study, cilostazol enhanced EPC mobilization and recruitment to the arterial injury sites and subsequently accelerated endothelial regeneration [25]. There are limited data regarding the interaction between pharmacological effect of cilostazol and EPC levels in high-risk patients; therefore, we conducted this study to evaluate the EPC-mobilizing effect of adjunctive cilostazol therapy in AMI patients.

## 2. Methods

This EPISODE (Endothelial Progenitor cell mobilization in AMI patients with ciloStazOl aDdEd) study was a single-center, prospective, randomized, placebo-controlled, double-blind trial (Figure 1). The study protocol and the informed consent form were approved by the institutional review boards of the hospital (GNUHIRB 2012-01-005). Written informed consent was obtained from all patients, and the study was performed in accordance with the Good Clinical Practice Guidelines and the principles of the Declaration of Helsinki.

### 2.1. Patients

Patients were eligible if they were diagnosed with naïve AMI and undergoing successful coronary stent implantation. Major exclusion criteria were as follows: (1) high-risk patients for thrombotic event; (2) a history of active bleeding or bleeding diatheses; (3) contraindication to antiplatelet therapy; (4) hemodynamic or electrical instability; (5) oral anticoagulation therapy; (6) left ventricular ejection fraction <30%, (7) leukocyte count <3000/mm^3^ and/or platelet count <100,000/mm^3^; (8) aspartate aminotransferase or alanine aminotransferase >3 times the respective the upper limit; (9) serum creatinine level >3.5 mg/dL; (10) stroke within 3 months; (11) pregnancy; (12) noncardiac disease with a life expectancy <1 year; (13) any patients not tolerable or suitable for coronary intervention; and (14) inability to follow the protocol.

### 2.2. Study Design

Suspected patients with AMI were loaded with 600-mg clopidogrel and 300-mg aspirin in the emergency room. All PCI procedures were performed according to the standard technique [1]. Following intervention, all patients were treated with the recommended pharmacological therapy of dual antiplatelet therapy (DAPT) with aspirin (100 mg daily) and clopidogrel (75 mg daily). After 3 to 5 days post-PCI (before discharge), patients were randomly allocated to the cilostazol (CILO) group or the placebo group (1:1 fashion) based on a computer-generated randomization sequence. For the CILO group, cilostazol-SR 200 mg daily was added to the DAPT. In the Placebo group, placebo tablet was administered on top of DAPT. Any change in study medications was not permitted during the study period. At 1-month follow-up visit at the outpatient clinic, drug adherence and adverse events were assessed by the attending physician blinded to the study drug, based on the medical interview, pill counting, and a dedicated questionnaire [26].

For the baseline measurement of EPC count and hemostatic measurements, blood samples were obtained 2 to 6 h after the last dose of the study drug from the antecubital vein (3–5 days post-PCI). If patients showed an appropriate drug adherence at 30-day follow-up, blood sampling for follow-up EPC and hemostatic measurements were collected 2–6 h after the last study drug administration from the antecubital vein.

### 2.3. Measurement of Circulating Endothelial Progenitor Cell

Laboratory measurements were performed within 2 h of blood sampling. Peripheral blood mononuclear cells (PBMNCs) were isolated by density gradient centrifugation over Ficoll (Sigma, St. Louis, MO, USA) for 25 min at 2300 rpm and were washed three times in phosphate buffered saline. Cells (1 × 10^5^) of PBMNCs were incubated for staining with 10 μL of each antibody (fluorescein isothiocyanate (FITC)-conjugated anti-human CD133 and CD34 monoclonal antibodies, and phycoerythrin (PE)-conjugated anti-human KDR monoclonal antibody) for 20 min at 4 °C in dark (Figure 2). Anti-IgG1 and IgG2a were used as isotype controls. Cells were washed twice. For flow cytometry analysis, 1 × 10^4^ cells were acquired and scored with flow cytometry analyzer and software (Beckman Coulter FC500, Brea, CA, USA). We gated lymphocytes and monocytes, and then examined them to count the positive cells. Intra-assay and inter-assay coefficient of variations were 4% and 7%, respectively, with lower detection limit of 0% for CD34+KDR+ EPC. In terms of CD133+KDR+EPC, intra-assay and inter-assay coefficient of variations were 6% and 9%, respectively, with lower detection limit of 0%.

### 2.4. Hemostatic Measurements

**VerifyNow assay:** The VerifyNow P2Y12 assay (Accriva, San Diego, CA, USA) is a whole-blood, point-of care, turbidimetric-based optical detection assay designed to measure agonist-induced platelet aggregation [27,28]. Our previous report demonstrated the strong correlation between the VerifyNow assay and light transmittance aggregometry (0.653 ≤ r ≤ 0.718) [27]. Blood samples were collected in 3.2% citrate Vacuette tubes (Greiner Bio-One Vacuette North America, Inc., Monroe, NC, USA). The measurement protocol was followed by the manufacturer’s recommendation and the details were described elsewhere [26]. The VerifyNow P2Y12 cartridge consists of two channels: one channel contains fibrinogen-coated polystyrene beads, 20 μM adenosine diphosphate (ADP), and 22 nM prostaglandin E1; the optical signal of this channel is reported as P2Y12 Reaction Units (**PRU**). The second channel contains fibrinogen-coated polystyrene beads, 3.4 mM iso-thrombin receptor-activating peptide (protease-activated receptor (PAR)-1 agonist), and PAR-4-activating peptide. This channel was incorporated to estimate the maximal platelet function independent of P2Y_12_ receptor blockade (**BASE**).

**Thromboelastography (TEG):** Global haemostasis was assessed by the TEG 5000 global haemostasis assay (Hemonetics Corp, BrainTree, MS, USA) [29,30]. The TEG Hemostasis Analyzer with automated analytical software provides measurements of the viscoelastic properties of a clot. Blood samples were drawn into Vacutainer tubes containing 3.2% trisodium citrate (Becton-Dickinson, Franklin Lakes, NJ, USA). In brief, 500 μL citrate blood was mixed by inversion with kaolin, and 340 μL of activated blood was transferred to a reaction cup containing 20 μL of 200 mM of calcium chloride. The fixed pin is suspended in a vibrating cup containing a whole blood sample. As the thrombus is formed, the pin is linked to the cup. Pin movement is detected and recorded as an electrical signal. The degree of platelet contribution to clot strength through platelet-fibrin binding is directly related to magnitude of pin motion and amplitude of tracking. Kaolin-induced maximum amplitude (**MA_thrombin_**) represents maximum platelet-fibrin clot strength and is affected by changes in fibrinogen, platelet count and function. Intra-assay and total precisions were 9% and 10%, respectively, with a lower detection limit of 2 mm.

### 2.5. Endpoints

Primary endpoint was the relative change of EPC count (∆CD133^+^/KDR^+^ and ∆CD34^+^/KDR^+^) between baseline and 30-day measurements.
$$\Delta \mathrm{CD}133^{+}/\mathrm{KD}R^{+}=\frac{{\left(\mathrm{CD}133^{+}/\mathrm{KD}R^{+}\;\mathrm{per}\;10^{4}\mathrm{mononuclear}\;\mathrm{cells}\right)}_{\mathrm{follow}-\mathrm{up}}\;-\;{\left(\mathrm{CD}133^{+}/\mathrm{KD}R^{+}\;\mathrm{per}\;10^{4}\mathrm{mononuclear}\;\mathrm{cells}\right)}_{\mathrm{baseline}}}{{\left(\mathrm{CD}133^{+}/\mathrm{KD}R^{+}\;\mathrm{per}\;10^{4}\mathrm{mononuclear}\;\mathrm{cells}\right)}_{\mathrm{baseline}}}\;\times \;100\;(\%)$$
$$\Delta \mathrm{CD}34^{+}/\mathrm{KD}R^{+}=\frac{{\left(\mathrm{CD}34^{+}/\mathrm{KD}R^{+}\;\mathrm{per}\;10^{4}\mathrm{mononuclear}\;\mathrm{cells}\right)}_{\mathrm{follow}-\mathrm{up}}-{\left(\mathrm{CD}34^{+}/\mathrm{KD}R^{+}\;\mathrm{per}\;10^{4}\mathrm{mononuclear}\;\mathrm{cells}\right)}_{\mathrm{baseline}}}{{\left(\mathrm{CD}34^{+}/\mathrm{KD}R^{+}\mathrm{per}10^{4}\mathrm{mononuclear}\;\mathrm{cells}\right)}_{\mathrm{baseline}}}\times \;100\;(\%)$$

Secondary endpoints were: (1) PRU and BASE values at 30-day follow-up; (2) MA_thrombin_ values at 30-day follow-up; and (3) the correlation between the changes of EPC subsets and hemostatic measurements. In addition, ischemic events and any serious complication were evaluated for 30 days, and bleeding events were measured according to the Bleeding Academic Research Consortium (BARC) criteria [31].

### 2.6. Statistical Analysis

There have been no studies to evaluate the effect of cilostazol on EPC mobilization in patients with AMI. Therefore, the sample size calculation was based on a previous study evaluating the EPC-mobilizing effect of cilostazol versus aspirin in diabetic patients with cerebral ischemia [32]. After four-month treatment, the relative increase in EPC count was about 50% higher in the cilostazol users compared with the aspirin users (101.8% versus 50.6%). We assumed that adjunctive cilostazol to DAPT would increase EPC count by 25% compared with DAPT alone. Thus, at least 27 patients in each group were needed to detect a relative difference of 25% with a power of 95%, a two-sided α error = 0.05, and a standard deviation (SD) of 0.25. Considering a 10% drop-out rate, we enrolled 30 patients (PS program version 3.1.2). The Kolmogorov–Smirnov test was performed to analyze the normal distribution of continuous variables. Continuous variables were presented as means ± SD or as median (interquartile range (IQR)) as appropriate, while categorical variables are reported as frequencies and percentages. The Student unpaired t test for parametric continuous variables and the Mann–Whitney U test for nonparametric continuous variables were used. Paired t for parametric continuous variables and Wilcoxon Signed Rank test for nonparametric continuous variables were used for within group comparisons. Comparisons between categorical variables were performed using the Pearson Chi-square test or Fisher exact test, as appropriate. The correlations between the changes of EPC level and hemostatic measurements were evaluated with Pearson’s correlation test. A *p* value < 0.05 was considered statistically significant, and statistical analyses were performed using SPSSv24.0 software (SPSS Inc., Chicago, IL, USA).

## 3. Results

During the study period, no patients suffered from ischemic event, any serious complication (e.g., worsening heart failure), or serious bleeding event (BARC ≥ 2). Two patients in the CILO group discontinued the study drug due to headache, and one patient in the placebo group withdrew the participation of the study (Figure 1). Baseline characteristics were similar between the groups (Table 1 and Table 2). Especially, type and dose of statins were well balanced between the groups.

After 30-day follow-up, inflammatory markers (white blood cell counts, neutrophil-to-lymphocyte ratio, and high-sensitivity C-reactive protein) and lipid profile (triglyceride and low-density lipoprotein-cholesterol) were lowered (Table 2). However, there were no significant differences between the groups in terms of these measures.

### 3.1. Effects of Cilostazol on Circulating EPC Counts

The CILO versus placebo group showed higher ∆CD133^+^/KDR^+^ (difference 216%, 95% confidence interval [CI] 44 ~ 388%, *p* = 0.015) and ΔCD34^+^/KDR^+^ (difference 183%, 95% CI 25 ~ 342%, *p* = 0.024) (Figure 3).

At baseline, circulating EPC levels were similar between the groups (Table 3). At 30-day follow-up, the CILO group showed greater CD133^+^/KDR^+^ EPC per 10^4^ mononuclear cells compared with the placebo group (*p* = 0.014). The level of CD34^+^/KDR^+^ EPC per 10^4^ mononuclear cells at 30-day appeared to be higher in the CILO versus placebo group but did not reach the statistical significance (*p* = 0.108). The 30-day cilostazol administration significantly enhanced the levels of both EPC subsets (CD133^+^/KDR^+^ per 10^4^ mononuclear cells: 68 ± 78 to 267 ± 471, *p* = 0.018 and CD34^+^/KDR^+^ per 10^4^ mononuclear cells: 164 ± 225 to 388 ± 523, *p* = 0.029) but placebo administration did not (CD133^+^/KDR^+^ per 10^4^ mononuclear cells: 74 ± 116 to 44 ± 54, *p* = 0.192, and CD34^+^/KDR^+^ per 10^4^ mononuclear cells: 161 ± 210 to 188 ± 391, *p* = 0.537) (Figure 4).

### 3.2. Effects of Cilostazol on Hemostatic Measurements

There were no differences in baseline PRU values between the CILO versus Placebo group (185 ± 50 versus 187 ± 71 PRU, *p* = 0.857) (Table 3). After 30-day follow-up, the CILO group showed significantly lower PRUs compared with the Placebo group (130 ± 45 versus 169 ± 62 PRU; difference 39 PRU, 95% CI 10 ~ 68 PRU, *p* = 0.009) (Figure 5). At baseline and 30-day follow-up, BASE (VerifyNow) and MA_thrombin_ (TEG) values did not differ between the groups (Table 3).

### 3.3. Correlation between Changes of EPC Counts and Hemostatic Measurements

The correlations between the change of PRU and the changes of EPC counts were not significant (r = 0.058, *p* = 0.670 for ∆CD133^+^/KDR^+^ EPC versus ∆PRU; r = 0.140, *p* = 0.301 for ∆CD34^+^/KDR^+^ EPC versus ∆PRU, respectively) (Figure 6A). In addition, PRU levels at baseline and 30-day follow-up did not show any significant correlations with the changes of EPC subsets (all r values ≤ 0.105) (Figure 6B,C). In addition, the correlations between BASE (VerifyNow)/MA_thrombin_ (TEG) and EPC changes were insignificant (all r values ≤ 0.272 and ≤ 0.106, respectively) (figures not shown).

## 4. Discussion

The EPISODE study first demonstrated the influence of cilostazol on EPC mobilization in AMI patients. The key findings of the study are as below; (1) adjunctive cilostazol therapy in addition to DAPT significantly increased EPC levels compared with DAPT alone (as indicated by elevated levels of CD34^+^/KDR^+^ and CD133^+^/KDR^+^ EPC at 30 days in the cilostazol group); (2) triple antiplatelet therapy with cilostazol versus DAPT was associated with the reduction of PRUs; and (3) EPC mobilization by cilostazol may not be related with its effect of platelet inhibition. These findings suggest that cilostazol may have additional beneficial effects on platelets and endothelium after PCI in AMI patients.

Multiple studies demonstrated the benefits of adjunctive cilostazol to prevent short- and long-term atherothormbotic events after cardiovascular stenting [22,23,24]. Triple antiplatelet therapy comprising cilostazol and DAPT was more effective than conventional DAPT in reducing stent-related and stent-unrelated thrombotic events in high-risk patients including AMI. Cilostazol was also effective in reducing the risks of in-stent restenosis and repeated revascularization in these patients [22,23,24]. These clinical benefits of cilostazol have been attributed to its additional platelet inhibition on top of DAPT in PCI-treated patients. However, high on-clopidogrel platelet reactivity was more convincingly associated with the risk of stent thrombosis but less with in-stent restenosis [23,33,34]. EPC as a key effector of endothelial regeneration has been significantly associated with the risks of revascularization and thrombotic events after PCI [16]. Accordingly, EPC-mobilizing effect of cilostazol may be another important mechanism, independent of platelet inhibition, to prevent the ischemic events including in-stent restenosis after coronary stenting. Among various pathways, the adenosine-mediated pathway is one of the plausible mechanisms to explain EPC mobilization by cilostazol [35]. Cilostazol inhibits adenosine reuptake by erythrocytes, endothelial cells, muscle cells, and platelets at clinically relevant concentrations (3–5 μM) [36]. Increased plasma adenosine concentration enhances not only the antiplatelet and vasodilatory effects but also bone marrow-derived EPC mobilization [35,36]. Activation of adenosine A_2_A and A_3_ receptors directly modulates EPC migration. In addition, adenosine indirectly increases EPC level through stimulating proangiogenic molecules and subsequent engagement of pluripotent cells towards a proangiogenic state [37]. In addition, cilostazol is a clinically available phosphodiesterase-3 inhibitor, thereby increasing the cellar levels of cAMP, which promotes nitric oxide (NO) production by endothelial NO synthase (eNOS) phosphorylation and subsequent endothelial tube formation in endothelial cells [38]. eNOS phosphorylation also induces EPC mobilization in bone marrow and functional activation of EPC [39]. Through these mechanisms, cilostazol, as a dual blocker of cAMP and adenosine reuptake, may protect neointimal hyperplasia and target lesion revascularization after coronary stenting.

Numerous efforts have been made to elucidate optimal therapeutic strategies to increase EPC levels in patients with coronary artery disease [9,10,11,40]. Direct transplantation of exogenous EPC may be the most intuitive method to mobilize EPC [11,40]. However, the results are controversial regarding clinical benefits of EPC transplantation that can be attributed to poor engraftment and survival of transplanted cells [8]. The augmentation of endogenous EPCs at a site of injury is another possible method to increase EPC levels. Recently, an EPC-capturing stent has been proposed to harvest CD34^+^ EPC and subsequently facilitate the vascular healing process [8], but EPC-capturing stent treatment did not show convincing results [8,41]. Some pharmacological therapeutics showed promising results in increasing endogenous EPC after coronary stent implantation [8]. Statins are known to promote mobilization, migration, proliferation, and survival of EPC cells [11]. Indeed, high- versus moderate-intensity statin significantly increased the levels of EPC subsets (CD34^+^/KDR^+^ and CD133^+^/KDR^+^) and decreased in-stent neointima formation in diabetic patients undergoing DES implantation [42]. In this line, the EPISODE study added another therapeutic option of cilostazol on top of statin to enhance EPC mobilization in patients at high ischemic risks.

Recently, ticagrelor showed promising results in enhancing EPC levels and improving endothelial function after PCI [15,17]. Bonello et al. demonstrated that ticagrelor as compared to clopidogrel increased the number of EPC in ACS patients [15]. They suggested that ticagrelor may enhance EPC levels through increasing plasma adenosine concentration, that may be similar to cilostazol as shown in the current study [15]. Based on these results, pharmacological agents showing EPC-mobilizing effect (e.g., cilostazol or ticagrelor) may provide a therapeutic option to promote vascular repair and prevent thrombotic events after stent implantation. However, ticagrelor versus clopidogrel treatment significantly increased bleeding risk without a protective benefit against thrombotic events in Asian patients [18,19]. Considering the net clinical benefit without a trade-off of bleeding risk, triple antiplatelet therapy with cilostazol, clopidogrel, and aspirin may be a plausible regimen in ACS patients at high-bleeding risk (e.g., East Asians, elderly, and diabetes).

Our study has limitations as follows. First, the study was a single-center experience with a small sample size, but the study design (placebo-controlled, double-blind pattern) may overcome this limitation. Second, we investigated Asian patients only. Thus, generalizing these results to the Caucasian population might be limited. Third, the anti-inflammatory effect of cilostazol was not supported by the data of our study, which may be related to its limited potency against inflammation. Fourth, adjunctive use of cilostazol is not recommended by the Western guidelines due to the concern about worsening heart failure, but most clinical data from Asian countries did not suggest any serious complications in AMI patients. Finally, it would be very interesting to see if any future studies compare the effect of adjunctive cilostazol to DAPT versus DAPT with ticagrelor with respect to circulating EPCs in ACS patients.

In conclusion, this study showed that adjunctive use of cilostazol in addition to DAPT significantly increased EPC levels independent of platelet inhibition in AMI patients. Therefore, we suggest that adjunctive cilostazol has an additional pleiotropic effect of EPC mobilization in addition to platelet inhibition, which may be related to protective effects against atherothrombotic events after coronary stent implantation.

## Figures and Tables

**Figure 1 jcm-09-01678-f001:**
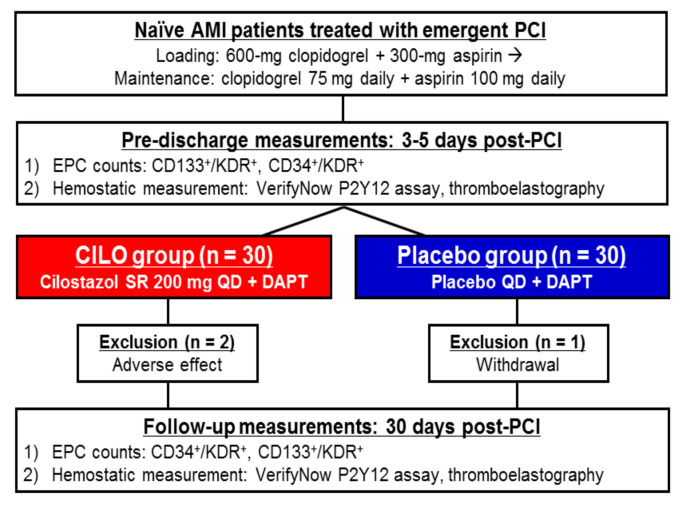
Study design. AMI = acute myocardial infarction; DAPT = dual antiplatelet therapy; EPC = endothelial progenitor cell; KDR = kinase insert domain-conjugating receptor; PCI = percutaneous coronary intervention.

**Figure 2 jcm-09-01678-f002:**
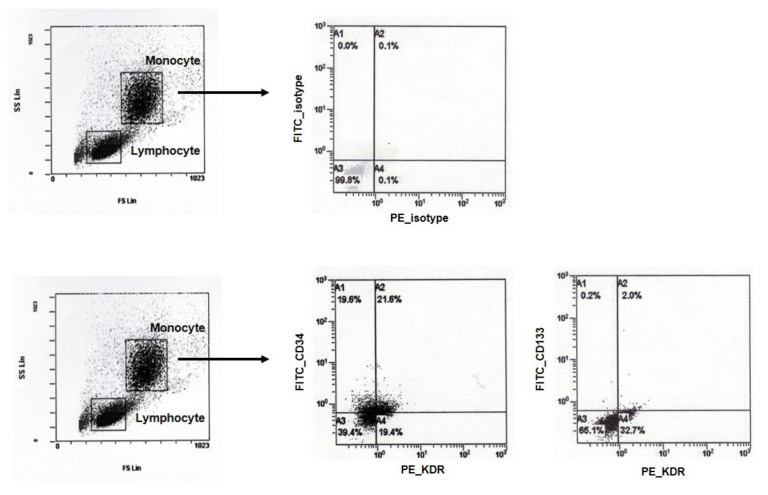
Measurement of circulating endothelial progenitor cells. Peripheral blood mononuclear cells (PBMNCs) were isolated by density gradient centrifugation over Ficoll. PBMNCs were analyzed for expression of CD34, CD133, and KDR. Quantitative analyses were performed by using flow cytometry. FITC = fluorescein isothiocyanate; PE = phycoerythrin.

**Figure 3 jcm-09-01678-f003:**
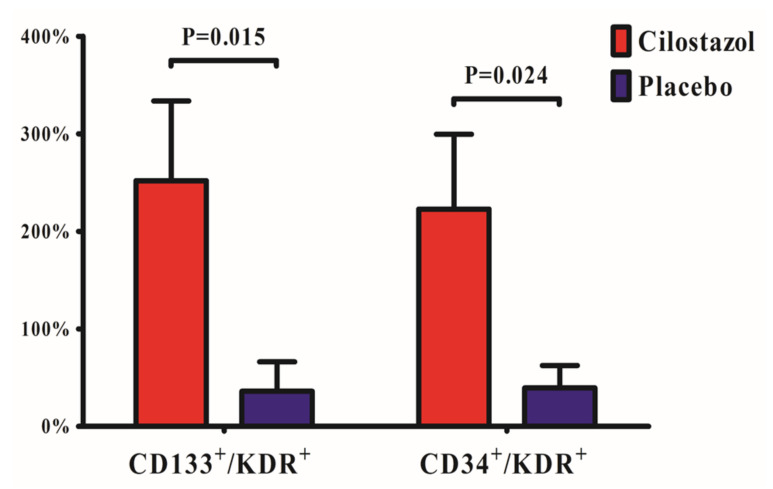
**Relative changes of endothelial progenitor cell counts between baseline and 30 days.** Cilostazol treatment was associated with significant relative changes of CD133^+^/KDR^+^ and CD34^+^/KDR^+^ endothelial progenitor cells compared with placebo. Data are expressed as the mean percentage of cells positive for each marker ± SEM.

**Figure 4 jcm-09-01678-f004:**
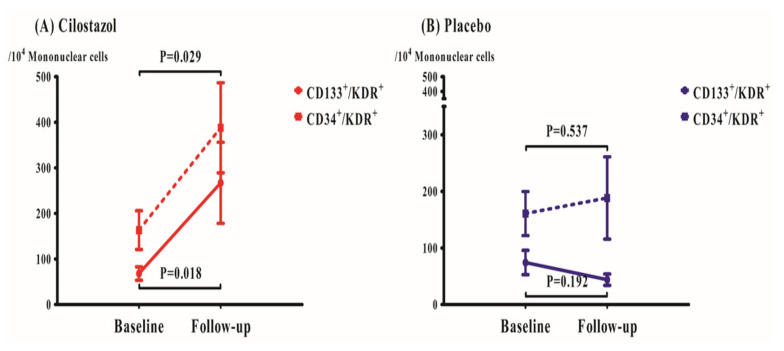
**Changes of endothelial progenitor cell counts at baseline and 30-day follow-up.** Cilostazol significantly increased both subsets of EPCs (**A**), whereas placebo had no significant effect on EPC levels (**B**). Data are expressed as the mean percentage of cells positive for each marker ± SEM.

**Figure 5 jcm-09-01678-f005:**
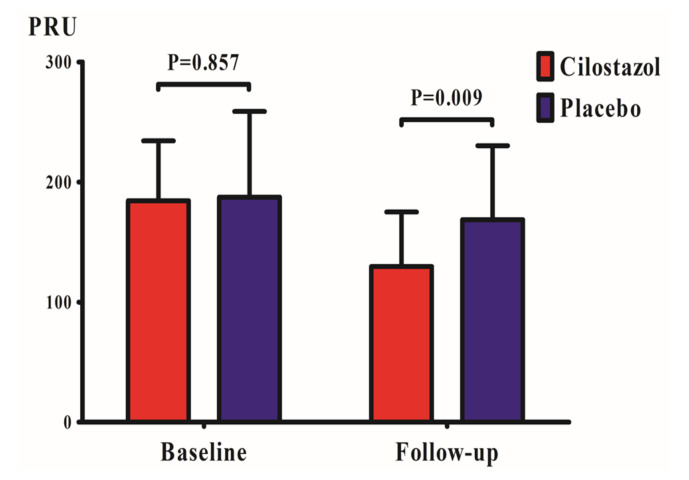
**Changes of platelet reactivity.** Adjunctive cilostazol on top of dual antiplatelet therapy showed a lower level of platelet reactivity compared with placebo. PRU = P2Y12 Reaction Units.

**Figure 6 jcm-09-01678-f006:**
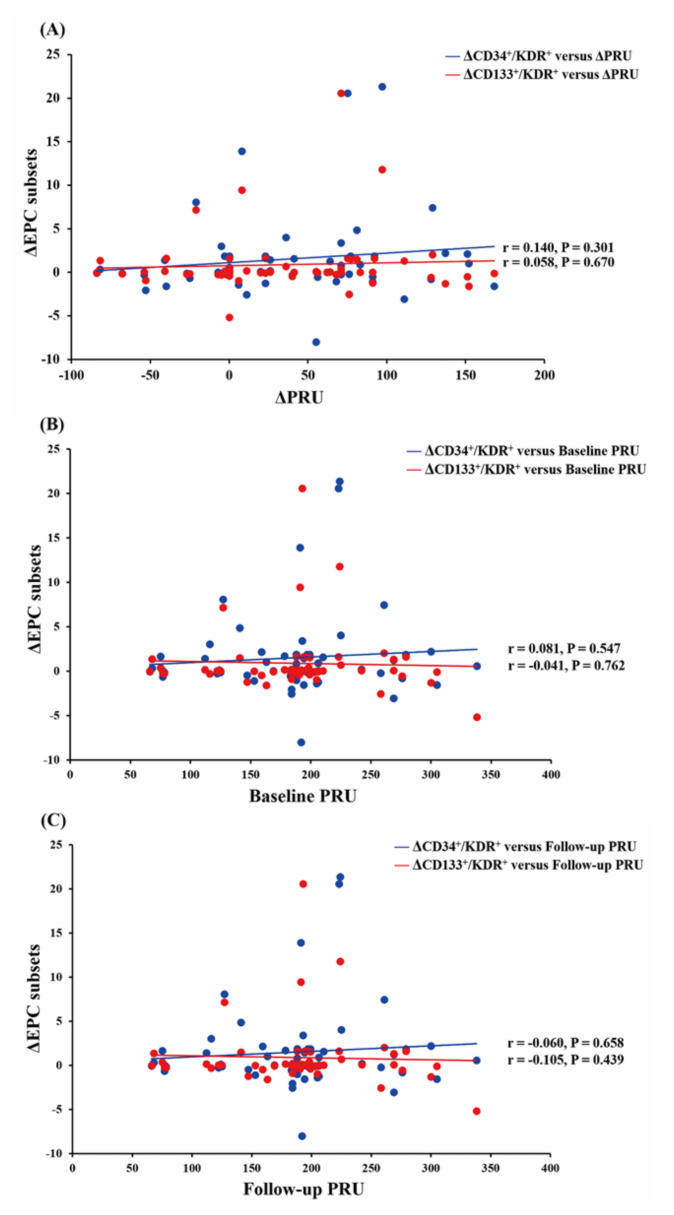
Correlation between EPC counts and platelet reactivity. Change of EPC levels did not show no significant correlation with change of platelet reactivity (**A**), and change of EPC levels were not related with the levels of platelet reactivity (**B** and **C**).

**Table 1 jcm-09-01678-t001:** Baseline demographic and procedural characteristics.

Variables	CILO Group (*n* = 28)	Placebo Group (*n* = 29)	*p* Value
Age, years	59.4 ± 11.3	60.8 ± 10.5	0.638
Male	24 (85.7)	24 (82.8)	0.760
BMI, kg/m^2^	25.2 ± 2.8	24.5 ± 2.0	0.251
Index presentation: STEMI, *n* (%)	17 (60.7)	17 (58.6)	0.872
**Risk factors, *n* (%)**			
Hypertension	16 (57.1)	21 (72.4)	0.227
Diabetes mellitus	6 (21.4)	7 (24.1)	0.807
Dyslipidemia	11 (39.3)	12 (41.4)	0.872
Chronic kidney disease	10 (35.7)	8 (27.6)	0.509
Current smoking	15 (53.6)	14 (48.3)	0.265
**Medications at discharge, n (%)**			
Aspirin	28 (100)	29 (100)	1.000
Clopidogrel	28 (100)	29 (100)	1.000
Beta blocker	21 (75.0)	25 (86.2)	0.284
Angiotensin antagonist	24 (75.0)	25 (86.2)	0.957
Calcium channel blocker	0 (0)	1 (3.4)	1.000
Statin	27 (96.4)	28 (96.6)	0.927
40-mg atorvastatin	18 (64.3)	19 (65.5)	
20-mg rosuvastatin	9 (32.1)	9 (31.0)	
Proton pump inhibitor	10 (35.7)	8 (27.6)	0.509
**Procedural characteristics**			
Pre-PCI TIMI 0–1 flow, n (%)	18 (64.3)	17 (58.6)	0.843
Post-PCI TIMI 3 flow, n (%)	27 (96.4)	27 (93.1)	1.000
Multivessel disease, n (%)	8 (28.6)	9 (31.0)	0.476
Multivessel PCI, n (%)	2 (7.1)	1 (3.4)	0.611
Aspiration thrombectomy, n (%)	16 (57.1)	12 (41.4)	0.234
IVUS usage	28 (100)	27 (93.1)	0.491
Infarct-related artery, n (%)			0.870
Left anterior descending	9 (32.1)	11 (37.9)	
Left circumflex	6 (21.4)	4 (13.8)	
Right coronary	13 (46.4)	14 (48.3)	
Intervention method, n (%)			0.635
Drug-eluting stent	27 (96.4)	28 (96.6)	
Bare metal stent	0 (0)	1 (3.4)	
Drug-coating balloon	1 (3.6)	0 (0)	
Stents number	1.1 ± 0.4	1.2 ± 0.5	0.577

BMI = body mass index; IVUS = intravascular ultrasound; PCI = percutaneous coronary intervention; STEMI = ST-segment elevation myocardial infarction; TIMI = Thrombolysis in Myocardial Infarction.

**Table 2 jcm-09-01678-t002:** Laboratory data.

Variables	CILO Group (*n* = 28)	Placebo Group (*n* = 29)	*p* Value
**Baseline measures (In-hospital)**			
WBC, × 10^3^/mm^3^	7.8 ± 1.8	8.1 ± 2.1	0.491
Neutrophil-to-lymphocyte ratio	2.35 ± 0.82	2.50 ± 1.23	0.596
Hemoglobin, g/dL	13.7 ± 1.7	13.2 ± 1.5	0.251
Platelets, × 10^3^/mm^3^	228.7 ± 33.0	249.6 ± 64.4	0.435
GFR, mL/min/1.73 m^2^ (MDRD)	88.0 ± 22.1	97.7 ± 15.6	0.060
Total cholesterol, mg/dL	167.7 ± 35.0	172.9 ± 36.5	0.587
Triglyceride, mg/dL	183.5 ± 91.0	183.8 ± 46.1	0.993
HDL cholesterol, mg/dL	44.7 ± 13.9	42.9 ± 12.0	0.606
LDL cholesterol, mg/dL	133.6 ± 41.0	123.5 ± 36.5	0.332
hs-CRP, mg/L	2.6 ± 3.7	4.5 ± 7.56	0.252
BNP, pg/mL	86 ± 120	76 ± 79	0.693
LV ejection fraction, %	55.3 ± 6.3	55.5 ± 6.8	0.903
HbA1c, %	6.6 ± 1.6	6.6 ± 1.7	0.891
Peak troponin-I, ng/mL	20.3 ± 8.5	21.6 ± 6.6	0.525
**Follow-up measures (1-month)**			
WBC, × 10^3^/mm^3^	6.9 ± 1.9	6.5 ± 1.8	0.402
Neutrophil-to-lymphocyte ratio	1.95 ± 0.82	2.07 ± 1.10	0.654
Hemoglobin, g/dL	13.8 ± 1.5	13.6 ± 1.6	0.662
Platelets, × 10^3^/mm^3^	240.9 ± 53.7	261.6 ± 68.9	0.575
GFR, mL/min/1.73 m^2^ (MDRD)	81.3 ± 25.4	89.4 ± 16.7	0.157
Total cholesterol, mg/dL	137.1 ± 23.3	142.4 ± 32.9	0.490
Triglyceride, mg/dL	120.5 ± 61.7	139.9 ± 73.8	0.291
HDL cholesterol, mg/dL	45.4 ± 10.5	42.1 ± 9.3	0.220
LDL cholesterol, mg/dL	80.3 ± 21.9	86.4 ± 28.0	0.370
hs-CRP, mg/L	1.9 ± 2.6	2.0 ± 2.0	0.907
BNP, pg/mL	96 ± 108	129 ± 120	0.154
LV ejection fraction, %	56.6 ± 6.2	57.5 ± 5.5	0.541

BNP = B-type natriuretic peptide; GFR = glomerular filtration rate; HbA1_C_ = hemoglobin A1_C_; HDL = high-density lipoprotein; hs-CRP = high-sensitivity C-reactive protein; LDL = low-density lipoprotein; LV = left ventricular; MDRD = Modification of Diet in Renal Disease; WBC = white blood cell.

**Table 3 jcm-09-01678-t003:** Circulating EPC levels and hemostatic measurements at baseline and follow-up.

Parameters	Baseline			Follow-Up		
	Cilostazol	Placebo	*p* Value	Cilostazol	Placebo	*p* Value
**Circulating EPC levels**						
CD133^+^/KDR^+^ per 10^4^ mononuclear cells	68 ± 78	74 ± 116	0.815	267 ± 471	44 ± 54	0.014
CD133^+^/KDR^+^ (%)	0.5 ± 0.6	0.8 ± 1.2	0.225	2.7 ± 4.7	0.5 ± 0.7	0.019
CD34^+^/KDR^+^ per 10^4^ mononuclear cells	164 ± 225	161 ± 210	0.963	388 ± 523	188 ± 391	0.108
CD34^+^/KDR^+^ (%)	1.7 ± 2.3	1.4 ± 1.2	0.618	3.9 ± 5.2	2.1 ± 3.9	0.149
**VerifyNow P2Y12 assay**						
PRU	185 ± 50	187 ± 71	0.857	130 ± 45	169 ± 62	0.009
BASE	258 ± 37	269 ± 48	0.427	231 ± 39	252 ± 51	0.190
**Thromboelastography**						
MA_thrombin_, mm	69.0 ± 9.3	70.8 ± 6.6	0.401	68.6 ± 14.2	68.5 ± 8.8	0.968

EPC = endothelial progenitor cell; MA = maximal amplitude; KDR = kinase insert domain-conjugating receptor; PRU = P2Y12 reaction unit.

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
