# Peer review of "Adjunctive Cilostazol to Dual Antiplatelet Therapy to Enhance Mobilization of Endothelial Progenitor Cell in Patients with Acute Myocardial Infarction: A Randomized, Placebo-Controlled EPISODE Trial"

_jcm, 2020, doi:10.3390/jcm9061678_

Round 1
Reviewer 1 Report
The manuscript by Park and co-authors entitled “Adjunctive cilostazol to dual antiplatelet therapy to enhance mobilization of endothelial progenitor cell in patients with acute myocardial infarction: A randomized, placebo-controlled ACCEL-EPISODE trial” assessed the effect of adjunctive cilostazol - a dual inhibitor of phosphodiesterase 3 and adenosine reuptake - on endothelial progenitor cell (EPC) mobilization and platelet reactivity in patients with acute myocardial infarction (AMI) following percutaneous coronary intervention. 60 Patients were included in this prospective, randomized clinical trial and randomized into two groups of 30 patients. Dual antiplatelet therapy was performed with aspirin and clopidogrel. The control group received placebo. The authors declare that adjunctive cilostazol, but not placebo, is associated with increased EPC mobilization and decreased platelet activation in AMI patients. They therefore propose that Cilostazol may prevent atherothrombotic events after coronary stent implantation.
Comments to the authors:
- 3, 2.2. Study design: Follow-up procedure should be explained more in detail. Has the attending physician been blinded? Were patients visited at home? Has the questionnaire been standardized and what was the content? Inaccuracies in follow-up assessment may easily influence the results and may thus lead to misinterpretation.
- 4/5, 2.4. Hemostatic measurements: The primary endpoint of this trial was the relative change of EPC level between baseline and day 30. EPC levels were analyzed by flow cytometry and hemostatic measurements were checked by VerifyNow and thromboelastography. Unfortunately, I am not familiar with the VerifyNow method and certainly there are others that do not know it as well. However, the interpretation of the results substantially depends on the measurement performance of this method. The authors therefore should mention potential limitations of VerifyNow (and for thromboelastography as well) and briefly state reliable validation data.
Author Response
Response to Comments from Reviewer 1
The manuscript by Park and co-authors entitled “Adjunctive cilostazol to dual antiplatelet therapy to enhance mobilization of endothelial progenitor cell in patients with acute myocardial infarction: A randomized, placebo-controlled ACCEL-EPISODE trial” assessed the effect of adjunctive cilostazol - a dual inhibitor of phosphodiesterase 3 and adenosine reuptake - on endothelial progenitor cell (EPC) mobilization and platelet reactivity in patients with acute myocardial infarction (AMI) following percutaneous coronary intervention. 60 patients were included in this prospective, randomized clinical trial and randomized into two groups of 30 patients. Dual antiplatelet therapy was performed with aspirin and clopidogrel. The control group received placebo. The authors declare that adjunctive cilostazol, but not placebo, is associated with increased EPC mobilization and decreased platelet activation in AMI patients. They therefore propose that cilostazol may prevent atherothrombotic events after coronary stent implantation.
We thank the reviewer’s for the comments, recognizing the strengths of our study. The points of concern are addressed in detail below, as presented in a significantly revised version of the manuscript.
- 2.2. Study design: Follow-up procedure should be explained more in detail. Has the attending physician been blinded? Were patients visited at home? Has the questionnaire been standardized and what was the content? Inaccuracies in follow-up assessment may easily influence the results and may thus lead to misinterpretation.
>>> Response: Thanks for the reviewer’s comment. This ACCEL-EPISODE study was conducted in the prospective, randomized, placebo-controlled, double-blind manner. Therefore, the attending physician did not know the component of the study drug till the end of study. In Korea, PCI-treated patients routinely visit the outpatient clinic where PCI procedure was performed. Prevalence of BARC bleeding was evaluated with a validated questionnaire (Serebruany V, et al. Eur Heart J. 2010;31:227-35) and our group also published the research paper using this questionnaire (Kwon TJ, et al. Thromb Haemost. 2016;115:979-92.).
• Change of the manuscript, Methods, page 3:
At 1-month follow-up visit, drug adherence and adverse events were assessed by the attending physician, based on the medical interview, pill counting, and a dedicated questionnaire. à
At 1-month follow-up visit at the outpatient clinic, drug adherence and adverse events were assessed by the attending physician blinded to the study drug, based on the medical interview, pill counting, and a dedicated questionnaire [25].
• Addition of the manuscript, References, page 13:
26. Serebruany V, Rao SV, Silva MA, et al. Correlation of inhibition of platelet aggregation after clopidogrel with postdischarge bleeding events: assessment by different bleeding classifications. Eur Heart J 2010;31:227-235.
- 2.4. Hemostatic measurements: The primary endpoint of this trial was the relative change of EPC level between baseline and day 30. EPC levels were analyzed by flow cytometry and hemostatic measurements were checked by VerifyNow and thromboelastography. Unfortunately, I am not familiar with the VerifyNow method and certainly there are others that do not know it as well. However, the interpretation of the results substantially depends on the measurement performance of this method. The authors therefore should mention potential limitations of VerifyNow (and for thromboelastography as well), and briefly state reliable validation data.
>>> Response: Thanks to the reviewer’s comments. Although light transmittance aggregometry (LTA) has been considered as the standard method of platelet function test, there are numerous limitations such as complex process requiring the dedicated personnel and wide CVs according to the laboratory setting. Severe point-of-care platelet function tests have been developed to overcome the limitation of LTA. VerifyNow test is the best validated method and numerous clinical data have suggested the clinical impact of VerifyNow in patients with ischemic heart disease. Our group also published the validation data between LTA and VerifyNow in terms of laboratory correlation and clinical outcome (Kim IS, et al. J Thromb Thrombolysis 2010;30:486-495.). The expert consensus strongly supported the clinical use of platelet function test, mostly based on the result of VerifyNow test (Tantry US, et al. J Am Coll Cardiol. 2013;62:2261-73.). Our group published about 30 SCI papers using VerifyNow test. In addition, TEG system has been used to measure global hemostatic process over 50 years. This assay was validated by numerous experimental data and our group published about 10 SCI papers using TEG system. Based on the reviewer’s recommendation, we added the validation data and consensus document for the readership.
• Addition of the manuscript, References, pages 13-14:
27.Kim IS, Jeong YH, Kang MK, et al. Correlation of high post-treatment platelet reactivity assessed by light transmittance aggregometry and the VerifyNow P2Y12 assay. J Thromb Thrombolysis 2010;30: 486-495.
28. Tantry US, Bonello L, Aradi D, et al.; Working Group on On-Treatment Platelet Reactivity. Consensus and update on the definition of on-treatment platelet reactivity to adenosine diphosphate associated with ischemia and bleeding. J Am Coll Cardiol 2013;62:2261-2273.
30. Jeong YH, Bliden KP, Antonino MJ, Tantry US, Gurbel PA. Usefulness of thrombelastography platelet mapping assay to measure the antiplatelet effect of P2Y12 receptor inhibitors and high on-treatment platelet reactivity. Platelets 2013;24:166-169.
Reviewer 2 Report
It was a great honour to review the manuscript “Adjunctive cilostazol to dual antiplatelet therapy to enhance mobilization of endothelial progenitor cell in patients with acute myocardial infarction: A randomized, placebo-controlled ACCEL-EPISODE trial” by Park et al.
Please find my comments below.
Abstract:
Line 19/20: Incomplete sentence. Protect against what?
Introduction:
Line 45: What do you mean with unrelated ischaemic events? TIA? CVA?
Line 46: Crooked sentence, please rephrase. Furthermore I think that event rates after pPCI for STEMI are not that high.
Line 57-58-59 Crooked sentence. Please rephrase
Methods:
Please define AMI. Were al patients admitted with a STEMI? Or was this a heterogeneous group of STEMI and NSTEMI?
Why were patients with a AMI loaded with Clopidogrel? According to the guidelines first choice is Ticagrelor. Were these patients at high bleeding risk? If so, why were they not excluded?
Why were there no thrombo-embolic events included in the secondary endpoints?
Results:
Why start the results with the secondary endpoints? Why not with the main findings? And why were there no ischaemic events if you don’t mention measuring these in your methods.
Why did almost half of the patients undergo aspiration thrombectomy? Exclusion criteria included high thrombus load (visible thrombus). Also, standard thrombus aspiration is no longer recommended.
Why was a BMS used?
Discussion:
The syntax is makes this manuscript somewhat difficult to read. Please consult a native-speaker to resolve this. Otherwise no comments
Conclusion:
Somewhat bold conclusion. You did not study atherothrombotic events after coronary stent implantation. The sample size is also too small for that. You need a larger RCT for that.
Author Response
Response to Comments from Reviewer 2
It was a great honour to review the manuscript “Adjunctive cilostazol to dual antiplatelet therapy to enhance mobilization of endothelial progenitor cell in patients with acute myocardial infarction: A randomized, placebo-controlled ACCEL-EPISODE trial” by Park et al.
We thank the reviewer’s for the comments, recognizing the strengths of our study. The points of concern are addressed in detail below, as presented in a significantly revised version of the manuscript.
- Abstract: Line 18/19: Incomplete sentence. Protect against what?
>>> Response: Thanks for the reviewer’s comment. We revised the sentence.
• Change of the manuscript, Abstract, page 1:
Endothelial progenitor cells (EPCs) have shown the potential to protect against atherothrombotic event occurrences.
- Introduction:
1) Line 46: What do you mean with unrelated ischaemic events? TIA? CVA?
>>> Response: As the reviewer indicated, DAPT strategy in PCI-treated MI patients can reduce the risk of cerebrovascular event. To state clearly, we revised the sentence.
2) Line 47: Crooked sentence, please rephrase. Furthermore, I think that event rates after pPCI for STEMI are not that high.
>>> Response: Classic RCTs (PLATO & TRITON) including ticagrelor and prasugrel have shown about 10% of ischemic events during 1 year. A recent GLOBAL-LEADER trial showed about 5% of a composite of death, MI, stroke and stent thrombosis during 24 months. Under the development of system, procedure and medical treatment, MACE risk following MI has been decreased. However, the realistic nationwide clinical data including MI patients still showed consistently maintained ischemic risk (up to 10% during 1 year). Based on the reviewer’s comment, we changed the word.
3) Line 58-59 Crooked sentence. Please rephrase.
>>> Response: Based on the reviewer’s comment, we tried to explain the potential role of EPCs in vascular repair.
- Methods:
1) Please define AMI. Were all patients admitted with a STEMI? Or was this a heterogeneous group of STEMI and NSTEMI?
>>> Response: As Table 1 indicated, about 60% of the patients were presented with STEMI, which showed that the cohort consisted of heterogeneous group. However, proportion of STEMI and NSTEMI was almost similar between the two study arms.
2) Why were patients with a AMI loaded with Clopidogrel? According to the guidelines first choice is Ticagrelor. Were these patients at high bleeding risk? If so, why were they not excluded?
>>> Response: This kind of practice pattern would be related with the unique clinical feature in East Asian countries. In Korea, potent P2Y12 inhibitors have been fully reimbursed from 2015. Even though Western guidelines recommended the initial use of potent P2Y12 inhibitor in AMI patients, there is a crucial concern of increased bleeding risk in East Asian patients during potent P2Y12 inhibitor (“East Asian Paradox”) and many hospitals still use clopidogrel as the first-line treatment of AMI. In our center, potent P2Y12 inhibitors have been established as the default strategy of AMI patients from the end of 2017. In addition, the ACCEL-EPISODE trial enrolled AMI patients mostly between 2016 and 2017.
3) Why were there no thrombo-embolic events included in the secondary endpoints?
>>>Response: As the reviewer indicated already, the sample size (30 patients per a group) was too small to conclude the risk of thrombo-embolic events. We also checked the prevalence of ischemic events and any serious complication during 30-day follow-up.
- Results:
1) Why start the results with the secondary endpoints? Why not with the main findings? And why were there no ischaemic events if you don’t mention measuring these in your methods.
>>> Response: In this ACCEL-EPISODE trial, secondary endpoints were: 1) PRU and BASE values at 30-day follow-up; 2) MAthrombin values at 30-day follow-up; and 3) the correlation between the changes of EPC subsets and hemostatic measurements. The first two sentences were used to describe the patients’ characteristics, babseline laboratory measurements and the drop-out reason between the two groups. In the section of 3.1. Effects of cilostazol on circulating EPC counts, we changed the sequence of the sentences to show the primary endpoint at the first.
2) Why did almost half of the patients undergo aspiration thrombectomy? Exclusion criteria included high thrombus load (visible thrombus). Also, standard thrombus aspiration is no longer recommended.
>>> Response: Thanks for the reviewer’s comment. We now know the limited benefit of routine aspiration thrombectomy in STEMI patients. In our cohort, STEMI patients consisted of about 60%, and about 75% of STEMI patients were treated with routine aspiration thrombectomy, which may be partly related with the attending physician’s preference. The study enrolled the patients between 2016 and 2017, and about 30-40% of STEMI patients currently have been treated with the initial aspiration thrombectomy according to the atheroma burden. High-risk patients for thrombotic event may indicate the subjects treated with very complex PCI and multiple CV risk factors and so on. We thus revised the sentence.
3) Why was a BMS used?
>>> Response: East Asian doctors have preferred procedure-based approach (PCI performance over 95% in Korean MI patients) and DES usage (over 98%) for AMI patients. We now know the current generation DES is superior to BMS in terms of clinical efficacy and safety in AMI patients. In our cohort, one patient was treated with BMS, which may be related with the attending physician’s preference and worry for clinical situation.
- Discussion: The syntax makes this manuscript somewhat difficult to read. Please consult a native-speaker to resolve this. Otherwise no comments.
>>> Response: We asked a native-speaker (Tantry US, PhD, USA) to resolve the bookish style of the sentence for the readership. Hope the reviewer would be happy with this rephrasing in the Discussion section (pages 10-12).
- Conclusion: Somewhat bold conclusion. You did not study atherothrombotic events after coronary stent implantation. The sample size is also too small for that. You need a larger RCT for that.
>>> Response: Based on the reviewer’s comment, we tried to revise the sentence using a lower tone.
Round 2
Reviewer 2 Report
Still some language issues. Please have a native speaker review the paper. Otherwise a good study for publication in the JCM.
Author Response
According to the reviewer's comment, we got another English editing from a native speaker (Paul A. Gurbel, MD). Hope this version can satisfy the concern from the reviewer.